# Effect of community-led delivery of HIV self-testing on HIV testing and antiretroviral therapy initiation in Malawi: A cluster-randomised trial

**Pitchaya P. Indravudh**[1,2]*, **Katherine Fielding**[3,4], **Moses K. Kumwenda**[2],
**Rebecca Nzawa**[2], **Richard Chilongosi**[5], **Nicola Desmond**[2,6], **Rose Nyirenda**[7],
**Melissa Neuman**[3], **Cheryl C. Johnson**[8,9], **Rachel Baggaley**[8], **Karin Hatzold**[10],
**Fern Terris-Prestholt**[1], **Elizabeth L. Corbett**[2,9]

1 Department of Global Health and Development, London School of Hygiene & Tropical Medicine, London,
United Kingdom, 2 Malawi-Liverpool-Wellcome Trust Clinical Research Programme, Blantyre, Malawi,
3 Department of Infectious Disease Epidemiology, London School of Hygiene & Tropical Medicine, London,
United Kingdom, 4 School of Public Health, University of the Witwatersrand, Johannesburg, South Africa,
5 Population Services International Malawi, Lilongwe, Malawi, 6 Department of International Public Health,
Liverpool School of Tropical Medicine, Liverpool, United Kingdom, 7 Department of HIV and AIDS, Ministry of
Health, Lilongwe, Malawi, 8 Global HIV, Hepatitis and Sexually Transmitted Infections Programmes, World
Health Organisation, Geneva, Switzerland, 9 Department of Clinical Research, London School of Hygiene &
Tropical Medicine, London, United Kingdom, 10 Population Services International, Washington, District of
Columbia, United States of America

* pitchaya.indravudh@lshtm.ac.uk

pmed.1003608

University, UNITED STATES

**Data Availability Statement:** Primary data are
available at https://doi.org/10.17037/DATA.
00001971.

## Abstract

### Background

Undiagnosed HIV infection remains substantial in key population subgroups including adolescents, older adults, and men, driving ongoing transmission in sub-Saharan Africa. We evaluated the impact, safety, and costs of community-led delivery of HIV self-testing (HIVST), aiming to increase HIV testing in underserved subgroups and stimulate demand for antiretroviral therapy (ART).

### Methods and findings

This cluster-randomised trial, conducted between October 2018 and July 2019, used restricted randomisation (1:1) to allocate 30 group village head clusters in Mangochi district, Malawi to the community-led HIVST intervention in addition to the standard of care (SOC) or the SOC alone. The intervention involved mobilising community health groups to lead the design and implementation of 7-day HIVST campaigns, with cluster residents (≥15 years) eligible for HIVST. The primary outcome compared lifetime HIV testing among adolescents (15 to 19 years) between arms. Secondary outcomes compared: recent HIV testing (in the last 3 months) among older adults (≥40 years) and men; cumulative 6-month incidence of ART initiation per 100,000 population; knowledge of the preventive benefits of HIV treatment; and HIV testing stigma. Outcomes were measured through a post-intervention survey and at neighboring health facilities. Analysis used intention-to-treat for cluster-level outcomes.

**Funding:** The study was funded by Unitaid, awarded to KH and ELC (PO#8477-0-600; https://unitaid.org/). ELC is also funded by the Wellcome Trust (WT091769; https://wellcome.ac.uk/). The funders had no role in study design, data collection and analysis, decision to publish, or preparation of the manuscript.

**Competing interests:** The authors have declared that no competing interests exist.

**Abbreviations:** ART, antiretroviral therapy; CHW, community health worker; HIVST, HIV self-testing; HTS, HIV testing services; RD, risk difference; SOC, standard of care; VMMC, voluntary medical male circumcision.

Community health groups delivered 24,316 oral fluid-based HIVST kits. The survey included 90.2% (3,960/4,388) of listed participants in the 15 community-led HIVST clusters and 89.2% (3,920/4,394) of listed participants in the 15 SOC clusters. Overall, the proportion of men was 39.0% (3,072/7,880). Most participants obtained primary-level education or below, were married, and reported a sexual partner. Lifetime HIV testing among adolescents was higher in the community-led HIVST arm (84.6%, 770/910) than the SOC arm (67.1%, 582/867; adjusted risk difference [RD] 15.2%, 95% CI 7.5% to 22.9%; $p < 0.001$), especially among 15 to 17 year olds and boys. Recent testing among older adults was also higher in the community-led HIVST arm (74.5%, 869/1,166) than the SOC arm (31.5%, 350/1,111; adjusted RD 42.1%, 95% CI 34.9% to 49.4%; $p < 0.001$). Similarly, the proportions of recently tested men were 74.6% (1,177/1,577) and 33.9% (507/1,495) in the community-led HIVST and SOC arms, respectively (adjusted RD 40.2%, 95% CI 32.9% to 47.4%; $p < 0.001$). Knowledge of HIV treatment benefits and HIV testing stigma showed no differences between arms. Cumulative incidence of ART initiation was respectively 305.3 and 226.1 per 100,000 population in the community-led HIVST and SOC arms (RD 72.3, 95% CI −36.2 to 180.8; $p = 0.18$). In post hoc analysis, ART initiations in the 3-month post-intervention period were higher in the community-led HIVST arm than the SOC arm (RD 97.7, 95% CI 33.4 to 162.1; $p = 0.004$). HIVST uptake was 74.7% (2,956/3,960), with few adverse events (0.6%, 18/2,955) and at US$5.70 per HIVST kit distributed. The main limitations include the use of self-reported HIV testing outcomes and lack of baseline measurement for the primary outcome.

## Conclusions

In this study, we found that community-led HIVST was effective, safe, and affordable, with population impact and coverage rapidly realised at low cost. This approach could enable community HIV testing in high HIV prevalence settings and demonstrates potential for economies of scale and scope.

## Trial registration

**Clinicaltrials.gov** NCT03541382.

## Author summary

### Why was the study done?

- Prevalence of undiagnosed HIV infection is significant in adolescents, older adults, and men, with alternative HIV testing strategies needed to reach underserved population subgroups.

- The aim of the study was to evaluate whether community-led delivery of HIV self-testing (HIVST) increased HIV testing in underserved population subgroups and antiretroviral therapy (ART) initiations, how safely, and with what costs.

## What did the researchers do and find?

- We conducted a cluster-randomised trial in a rural, high HIV prevalence area of Malawi, randomising group village head clusters to the community-led HIVST intervention or standard of care (SOC) arms.

- The intervention involved mobilising established community health groups to lead the design and implementation of 7-day HIVST campaigns in addition to the SOC, which primarily included facility-based HIV testing services. Outcomes were assessed through a post-intervention survey and at neighbouring health facilities.

- The survey included 7,880 participants. Following delivery of community-led HIVST, lifetime testing increased by 15.2% for adolescents and recent testing increased by 42.1% for older adults and by 40.2% for men. Cumulative incidence of ART initiation apparently increased 3 months post-intervention, with difference between arms of 97.7 residents treated per 100,000 population.

- Uptake of HIVST was 74.7%, with limited adverse events and at US$5.70 per HIVST kit distributed.

## What do these findings mean?

- Community-led HIVST rapidly achieved high impact on HIV testing and ART initiation and high coverage of HIVST at low cost.

- National HIV programmes in priority settings should consider community-led HIVST for periodic community HIV testing to achieve HIV elimination goals. This approach has potential for further economies of scale and scope, with community health groups widely established throughout sub-Saharan Africa and increasing availability of self-care products.

## Introduction

In 2018, approximately 1.7 million people were newly infected with HIV, with most cases in sub-Saharan Africa [1]. Regionally, almost one-fifth of people living with HIV were unaware of their HIV status [1]. Gaps remain more substantial among adolescents 15 to 19 years, older adults 40 years and above, and men [2]. While HIV incidence has been declining, undiagnosed HIV infection in these key population subgroups are drivers of ongoing transmission, impeding achievement of HIV elimination goals [1]. Routine HIV testing is a critical component of providing early diagnosis and treatment to reduce HIV-related morbidity and mortality and maximise HIV prevention benefits [3].

HIV testing services (HTS) are being provided within the context of declining undiagnosed HIV [4]. Most HTS are facility-based, though barriers including HIV-related stigma and discrimination, lack of convenience, and economic costs for clients have hindered uptake among underserved subgroups [5,6]. Community-based HTS can diagnose individuals at earlier stages of disease [7] and improve treatment and viral suppression when combined with convenient antiretroviral therapy (ART) services [8]. Despite their contributions, costs are higher for community-based HTS and global funding for community health programmes has been

declining [7,9]. More efficient and scalable community strategies are needed to reach and maintain universal HIV testing in HIV-prevalent populations.

Among the most promising approaches are community-led strategies for disease prevention and management, which aim to involve underserved communities in decision-making, align health programmes with community-specific needs and experiences, and expand and sustain coverage [10]. Prior studies have shown increased coverage and efficiency and improved health outcomes when communities participate in the design and implementation of health services [11–14]. Within HIV, community groups have led demand creation activities for HIV services [12,15]. Recent innovations in self-care technologies are now expanding the breadth of services that could be delivered by communities [16].

HIV self-testing (HIVST) is a recommended approach that can facilitate novel HIV testing strategies [17]. Previous studies have demonstrated the effectiveness of door-to-door distribution of HIVST kits on increased HIV testing in Malawi and Zambia [18,19]. Given the impact of vertical community-based HIVST, we evaluated community-led delivery of HIVST. Specifically, we investigated the impact, safety, and costs of mobilising community health groups to lead the design and implementation of 7-day HIVST campaigns, aiming to increase HIV testing in underserved subgroups and stimulate demand for ART in a rural, high HIV prevalence area of Malawi.

## Methods

### Design

We conducted a cluster-randomised trial and allocated 30 group village head clusters to the community-led HIVST intervention in addition to the standard of care (SOC) or the SOC alone (S1 CONSORT Checklist). A cluster-randomised design was used since the intervention was delivered at group village head level. The study aimed to determine whether the intervention increased the proportion of the population who tested for HIV at cluster level, focusing on adolescents, older adults, and men. The trial also assessed impacts on population-level ART initiation, knowledge of the preventive benefits of HIV treatment, and HIV testing stigma; adverse events; and costs. A detailed protocol was published separately [https://hivstar.lshtm. ac.uk/protocols/] [20].

### Setting and participants

Mangochi is a rural district bordering Lake Malawi and Mozambique with adult HIV prevalence of 10.1% (Fig 1) [21]. Group village heads hold customary authority over a group of villages. Government community health workers (CHWs) oversee provision of basic health services with community health action groups at the group village head level. Community volunteers, including village health committees, provide services at village level.

Group village head clusters serviced by 5 government primary health centres were assessed by the study team for eligibility. Clusters were defined according to the boundaries of the group village head catchment area. Inclusion criteria for clusters prioritised a minimum population of 2,000 residents, distance of at least 5 kilometres to the health facility, and geographical separation between clusters. The study team obtained verbal consent from group village heads for cluster enrolment.

### Randomisation

The 30 group village head clusters were randomised (1:1) to the community-led HIVST or SOC arm. Restricted randomisation was used to ensure balance between arms for key factors that could influence the effect of the intervention [22]. Restriction criteria included health

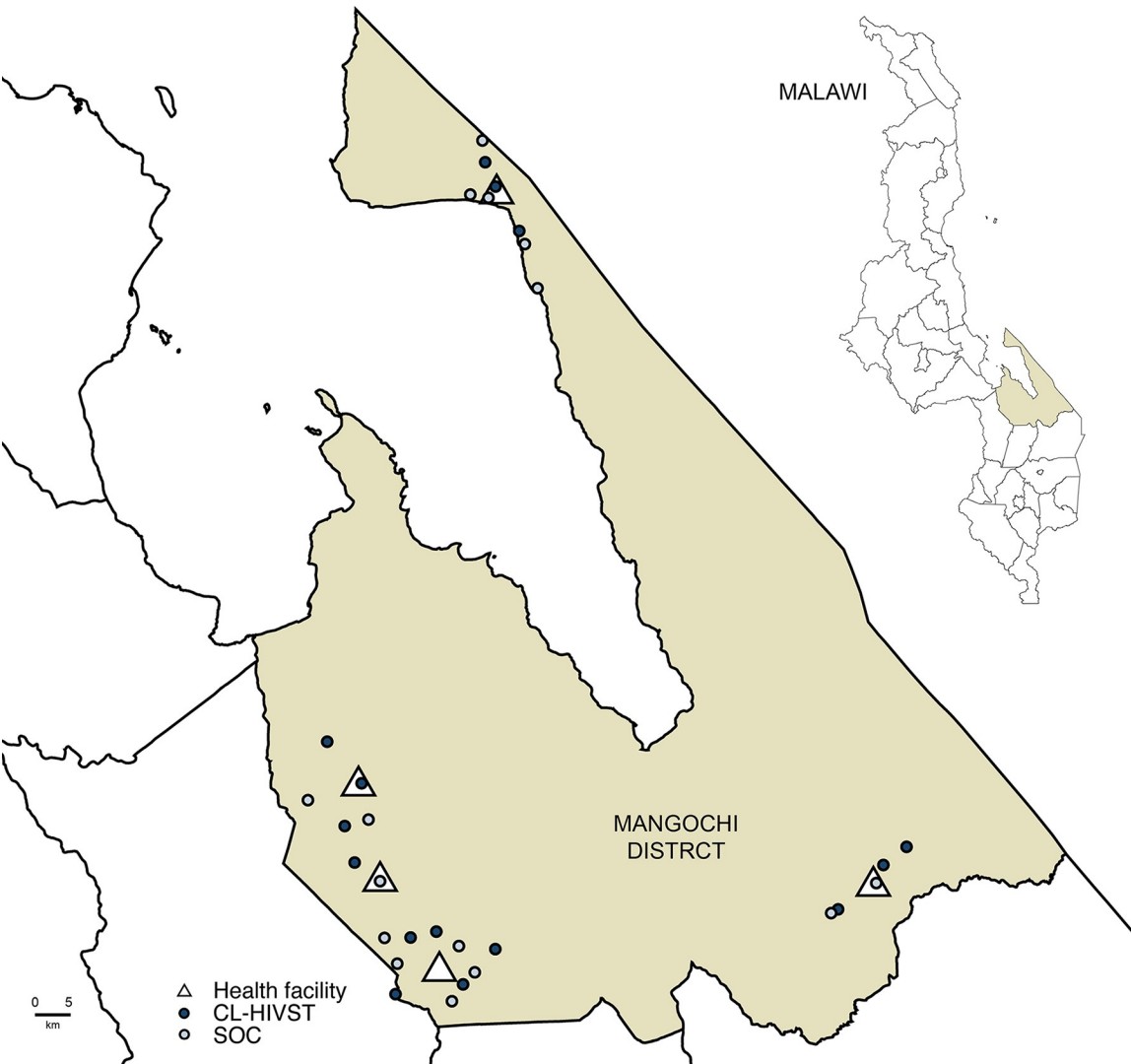

**Fig 1. Map of clusters in Mangochi district.** HIVST, HIV self-testing; SOC, standard of care. Map of Mangochi district with health facilities and group village head-defined clusters. Malawi National Spatial Data Centre, www.masdap.mw.

facility, population, distance from facility, and number of villages (S1 Text). From 12,540 unique combinations falling within the restriction parameters, PPI and KF drew a computer-generated random sample of 1,000 combinations, which were sequentially numbered.

On July 16, 2018, group village head clusters were randomised at a public ceremony with community and government representatives. Volunteers selected numbered balls corresponding to one combination and one arm allocation from an opaque bag. Masking of community implementers and residents was not feasible since the intervention was delivered at cluster level, but data were managed and analysed without reference to arm allocation where possible.

## Procedures

### Community-led HIV self-testing

The community-led HIVST intervention involved engaging established community health groups from 15 group village head clusters to lead the design and implementation of HIVST

campaigns in their areas. Implementation was staggered, with 2 to 3 clusters receiving the intervention every 14 days. Implementation was administered by the study team, including Population Services International Malawi, the Malawi-Liverpool-Wellcome Trust Clinical Research Programme, and the Ministry of Health. Formative research and piloting informed the design [20].

Following entrance meetings, the intervention proceeded in 3 stages, adapting participatory learning and action methods [23]. First, community health action groups and CHWs attended 2-day participatory workshops. Participants identified drivers of HIV infection, mapped HIV services and barriers to access, defined priority subgroups, and designed a 7-day HIVST campaign to be delivered in their areas. Specifically, participants planned strategies for distribution of HIVST kits, support for linkage to routine HIV care, demand creation for HIVST, social harms reporting, and monitoring and evaluation.

Second, community volunteers attended 2-day trainings on supporting use and interpretation of HIVST kits and providing information on linkage to routine HIV services, specifically confirmatory testing and ART initiation for reactive results, voluntary medical male circumcision (VMMC) for nonreactive results among men, and couples testing for serodiscordant results among partners. Volunteers were also trained on communication of HIV prevention messages, including effectiveness of ART, management of social harms, handling and storage of kits, and data collection.

Lastly, community volunteers delivered the campaign in their areas, supervised by community health action groups and CHWs. Implementation was based on strategies defined during participatory workshops for each cluster. HIVST sensitisation and distribution was conducted door-to-door and from fixed locations (e.g., schools, mosques, boreholes), social hotspots (e.g., fishing docks, sports fields, video shows), and community meetings. Support for linkage to routine services provided by community health groups included phone referrals and donated transportation funds. In addition to support provided by communities, the study team supplied the OraQuick HIV Self-Test (Orasure Technologies, Thailand), instructional materials, data collection tools, and nationally standardised gratuity of MWK 7,000 (US$10) per volunteer. HIVST kits could be taken by cluster residents aged 15 years and older. Residents could take an additional kit for secondary distribution and self-test with volunteer support or in private, with or without disclosing results.

## Standard of care

The SOC, which was also available in community-led HIVST clusters, included HIV testing available through the Ministry of Health. HTS are provided by lay counsellors at health facilities through periodic community-based outreach. HIV testing follows standard serial testing algorithms using finger-prick rapid diagnostic tests, with ART universally available immediately following an HIV-positive diagnosis.

## Outcomes and measurement

Outcomes were selected to understand the effect of the intervention on uptake of HIV services, especially among low-coverage subgroups. The primary outcome compared the proportion of adolescents (15 to 19 years) who self-reported lifetime testing for HIV between arms. Lifetime testing was a more relevant measure for adolescents since we anticipated that a high proportion of adolescents would have never tested [21], with the need for testing among this age group highly variable and dependent on the onset of sexual debut and HIV risk. We therefore hypothesised that the intervention would increase coverage of lifetime testing in a subgroup with limited previous experience of HIV testing, with a similar effect achieved on recent testing.

Secondary outcomes compared: self-reported recent HIV testing (in the last 3 months) among older adults (≥40 years) and men; cumulative 6-month incidence of ART initiation per 100,000 population; knowledge of the preventive benefits of HIV treatment; and HIV testing stigma. Exploratory outcomes compared: mutual knowledge of HIV status between sexual partners; recent HIV testing for adolescents; lifetime HIV testing for older adults and men; and HIV testing in the last 12 months for adolescents, older adults, and men. Cumulative incidence of VMMC was not assessed as specified in the protocol as services were discontinued prior to study initiation.

Outcomes were measured at cluster level through a post-intervention survey, except for ART initiations, which were captured at the 5 health facilities. The survey was administered 8 to 12 weeks after the intervention start in the community-led HIVST clusters or matched dates in the SOC clusters. Cluster residents were sampled for the survey to form the evaluation population. Within each cluster, villages with at least 500 residents and that included or was located near the group head village were randomly selected per cluster. In villages with approximately 500 residents, all households were eligible for the survey. In larger villages, 150 households were recruited in a clockwise spiral starting with the village head household, with multiple visits made to schedule interviews. Written informed consent or assent was obtained for residents aged 15 years and older in recruited households. Participants were interviewed on household and sociodemographic characteristics, HIV service use, and sexual behavior. A random sample (approximately 20%) received an HIV knowledge and attitudes module (Text A in S2 Text).

Clinic assistants at the 5 health facilities interviewed ART patients aged 15 years and older to establish cluster eligibility for 6 months following the intervention. Population estimates for cluster residents aged 15 years and older were obtained from village and facility registers and used as the denominator for cumulative incidence of ART initiations.

Process indicators measuring HIVST coverage were assessed through the survey and HIVST registers, which recorded sociodemographic information for residents collecting HIVST kits. Adverse events related to HIVST were captured through the survey and classified by severity [24].

Economic data on the total and unit costs of the intervention were collected from the provider perspective, with financial costs from expenditure records supplemented with full costs from direct observations and interviews (Text B in S2 Text). Costs are reported in 2018 US Dollars.

Planned outcome evaluation of a repeat community-led HIVST intervention [20] was not completed due to delayed study initiation, leaving insufficient time between post-intervention assessments. The study reported here had standalone outcomes and was not affected.

## Sample size

The study was powered to detect a 20% absolute difference between study arms in the primary outcome of lifetime HIV testing for adolescents [18]. We assumed 35% to 50% prevalence of testing among adolescents in the SOC arm based on national estimates [21]. Fifteen group village head clusters per arm with 50 adolescents of 250 residents per cluster provided 90% power at a 5% significance level. We assumed a coefficient of variation (k) of 0.25 based on guidelines for cluster-randomised trials [22]. The study was also powered to measure a difference in recent HIV testing in older adults and men and cumulative ART initiations (S1 Text).

## Statistical analysis

Analysis used intention-to-treat, that is participants within clusters were analysed based on cluster assignment to study arms rather than individual-level exposure to the intervention.

Outcomes were analysed at cluster level using established methods for cluster-randomised trials with a small number of clusters [22]. Specifically, risk differences, mean differences, and risk ratios for the intervention effect were calculated from cluster-level risks, means, and log risks, respectively (S1 Text). Cluster-level summaries were then compared between arms with a *t* test.

Using a 2-stage approach [22], effect estimates were adjusted for sex and age group a priori and any imbalance between arms in adolescent covariates. To estimate the risk difference and risk ratio, the first stage used logistic regression to adjust for confounding bias at the individual level. Predicted risks were then summed at cluster level and used to calculate the difference and ratio of observed and predicted values. A log transformation was applied to summaries as appropriate. The second stage used a *t* test to compare covariate-adjusted summary values between arms. To calculate the mean difference, similar procedures were applied using linear regression in the first stage.

A priori subgroup analysis compared the primary outcome by sex and age group (15 to 17 years, 18 to 19 years). Post hoc analysis compared cumulative incidence of ART initiation by first and last 3-month period. Statistical analysis used Stata version 14.0.

### Ethical considerations

The study is registered with ClinicalTrials.gov, NCT03541382. Ethical approvals were granted by the University of Malawi College of Medicine (P.01/18/2332), London School of Hygiene & Tropical Medicine (14761), and WHO (STAR-comm led CRT-Malawi). The study is part of the Unitaid/Population Services International HIV Self-Testing Africa Initiative (STAR) [http://hivstar.lshtm.ac.uk/].

### Results

The study population included 44,543 residents in 15 community-led HIVST clusters and 39,806 residents in 15 SOC clusters. The intervention was delivered from October 5, 2018 to January 17, 2019 by 157 community health action group members (cluster mean 10.5) and 190 community volunteers (cluster mean 12.7). Overall, 24,316 HIVST kits (cluster mean 1,621) were distributed, with 47.2% (*n* = 11,472) of kits distributed to men.

Outcomes were measured from December 5, 2018 to March 30, 2019 for the post-intervention survey and to July 31, 2019 for facility data collection. Fig 2 shows the trial flow diagram.

The survey included 90.2% (3,960/4,388) and 89.2% (3,920/4,394) of listed participants, respectively, in the community-led HIVST and SOC arms. Adolescent participation rates were similar at 90.2% (910/1,002) in the community-led HIVST arm and 86.4% (867/1,004) in the SOC arm.

Participant characteristics are summarised in Table 1. Overall, the proportion of men was 39.0% (3,072/7,880), which was below expected [21] with 84.6% (1,577/1,863) and 82.4% (1,495/1,814) responding in the community-led HIVST and SOC arms, respectively.

Most participants obtained primary-level education or below. The majority were married and reported a sexual partner. Characteristics were well balanced by arm, though differences in literacy, religion, ethnicity, and self-reported health status were observed for adolescents (Table 2).

### Primary and secondary outcomes

Lifetime HIV testing among adolescents was higher in the community-led HIVST arm (84.6%, 770/910) than the SOC arm (67.1%, 582/867), with adjusted risk difference (RD) of 15.2% (95% CI 7.5% to 22.9%; *p* < 0.001; Table 3 and Table A in S2 Text and Fig A in S2 Text). There was strong evidence that the effect of the intervention differed by age group (*p*-value for

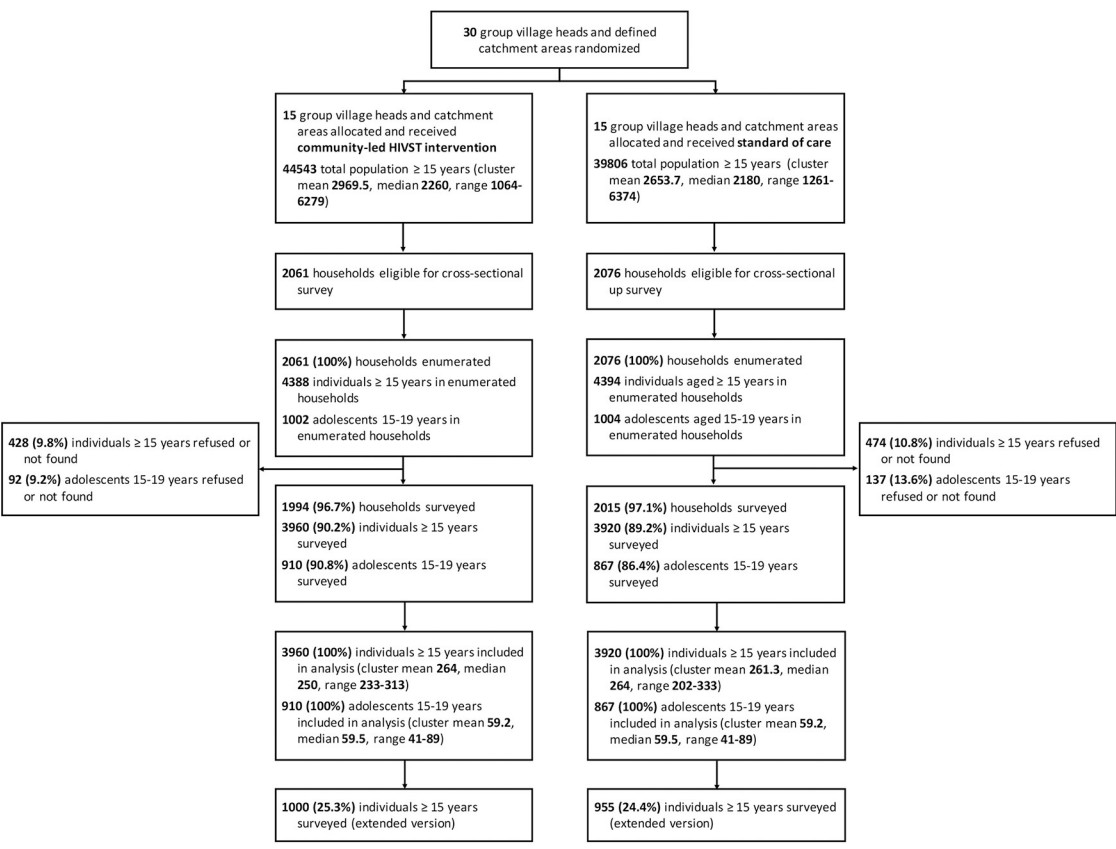

**Fig 2. Trial flow diagram.** Flow diagram of the cluster-randomised trial.

interaction = 0.02), with a more pronounced difference among 15 to 17 year olds (adjusted RD 21.5%, 95% CI 10.4% to 32.6%; $p < 0.001$) than 18 to 19 year olds (adjusted RD 10.8%, 95% CI 4.3% to 17.3%; $p = 0.002$). Lifetime testing was also higher for boys (adjusted RD 20.5%, 95% CI 10.7% to 30.3%; $p < 0.001$) than girls (adjusted RD 11.1%, 95% CI 2.8% to 19.4%; $p = 0.01$; $p$-value for interaction = 0.06).

Recent HIV testing (in the last 3 months) among older adults was higher in the community-led HIVST arm (74.5%, 869/1,166) than the SOC arm (31.5%, 350/1,111), with adjusted RD of 42.1% (95% CI 34.9% to 49.4%; $p < 0.001$). The proportion of recently tested men was 74.6% (1,177/1,577) in the community-led HIVST arm and 33.9% (507/1,495) in the SOC arm (adjusted RD 40.2%, 95% CI 32.9% to 47.4%; $p < 0.001$). Knowledge of the preventive benefits of HIV treatment and HIV testing stigma measures showed no differences between arms (Table 3).

Cumulative 6-month incidence of ART initiation was, respectively, 305.3 and 226.1 per 100,000 population in the community-led HIVST and SOC arms (RD 72.3, 95% CI −36.2 to 180.8; $p = 0.18$). In post hoc analysis, cumulative incidence in the 3-month post-intervention period was, respectively, 186.3 and 93.0 per 100,000 population in the community-led HIVST and SOC arms, with a larger effect in the first 3 months (RD 97.7, 95% CI 33.4 to 162.1; $p = 0.004$) than the last 3 months (RD −10.7, 95% CI −80.5 to 59.2; $p = 0.76$; $p$-value for interaction = 0.02).

In exploratory analyses, the intervention increased HIV testing in 3-month, 12-month, and lifetime periods, overall and among defined subgroups, and mutual knowledge of HIV status between sexual partners (adjusted RD 14.1%, 95% CI 8.6% to 19.5%; $p < 0.001$; Table A in S2 Text).

**Table 1. Comparison of population characteristics by study arm.**

| | Community-led HIVST | SOC |
|---|---|---|
| | n (%) | n (%) |
| **Household characteristics** | (N = 1,994) | (N = 2,015) |
| Adults (median/range)* | 2 (0–8) | 2 (0–10) |
| Children (median/range)* | 1 (0–1) | 1 (0–1) |
| Household wealth index† | | |
| Lowest | 368 (20.3%) | 341 (18.6%) |
| Second | 353 (19.4%) | 395 (21.6%) |
| Third | 361 (19.9%) | 362 (19.8%) |
| Fourth | 358 (19.7%) | 373 (20.4%) |
| Highest | 375 (20.7%) | 358 (19.6%) |
| **Individual characteristics** | (N = 3,960) | (N = 3,920) |
| Male | 1,577 (39.8%) | 1,495 (38.1%) |
| Age (median/range) | 29 (15–96) | 29 (15–98) |
| Age group | | |
| 15–19 years | 910 (23.0%) | 867 (22.1%) |
| 20–24 years | 631 (15.9%) | 675 (17.2%) |
| 25–39 years | 1,253 (31.6%) | 1,267 (32.3%) |
| ≥40 years | 1,166 (29.4%) | 1,111 (28.3%) |
| Marital status‡ | | |
| Married or living together | 2,428 (61.3%) | 2,467 (62.9%) |
| Separated, divorced, or widowed | 612 (15.5%) | 542 (13.8%) |
| Never married | 918 (23.2%) | 910 (23.2%) |
| Educational attainment§ | | |
| None | 1,730 (43.7%) | 1,764 (45.0%) |
| Primary | 1,902 (48.0%) | 1,838 (46.9%) |
| Secondary or higher | 328 (8.3%) | 317 (8.1%) |
| Literate‖ | 2,196 (55.5%) | 2,066 (52.7%) |
| Muslim | 2,840 (71.7%) | 3,008 (76.7%) |
| Ethnicity | | |
| Yao | 2,778 (70.2%) | 2,942 (75.1%) |
| Ngoni | 546 (13.8%) | 443 (11.3%) |
| Other | 636 (16.1%) | 535 (13.6%) |
| Resident in the last 2 months | 3,877 (97.9%) | 3,830 (97.7%) |
| Self-rated health status¶ | | |
| Very good | 1,546 (39.1%) | 1,314 (33.5%) |
| Good | 1,738 (43.9%) | 1,810 (46.2%) |
| Fair | 338 (8.5%) | 389 (9.9%) |
| Poor | 337 (8.5%) | 407 (10.4%) |
| Reported current sexual partner** | 2,875 (72.6%) | 2,931 (74.8%) |
| Circumcised (for men)†† | 1,335 (84.9%) | 1,285 (86.0%) |

HIVST, HIV self-testing; SOC, standard of care.

*32 missing values in the community-led HIVST arm and 8 missing values in the SOC arm.

†179 missing values in the community-led HIVST arm and 186 missing values in the SOC arm.

‡2 missing values in the community-led HIVST arm and 1 missing value in the SOC arm.

§1 missing value in the SOC arm.

‖1 missing value in the community-led HIVST arm.

¶1 missing value in the community-led HIVST arm.

**1 missing value in the SOC arm.

††5 missing values in the community-led HIVST arm.

**Table 2. Comparison of characteristics for adolescents (15–19 years) by study arm.**

|  | Community-led HIVST | SOC |
|---|---|---|
|  | n (%) | n (%) |
| **Individual characteristics** | (*N* = 910) | (*N* = 867) |
| Male | 387 (42.5%) | 381 (43.9%) |
| Age (median/range) | 18 (15–19) | 18 (15–19) |
| Age group |  |  |
| 15–17 years | 400 (44.0%) | 384 (44.3%) |
| 18–19 years | 510 (56.0%) | 483 (55.7%) |
| Marital status* |  |  |
| Married or living together | 138 (15.2%) | 147 (17.0%) |
| Separated, divorced, or widowed | 34 (3.7%) | 20 (2.3%) |
| Never married | 738 (81.1%) | 699 (80.7%) |
| Educational attainment |  |  |
| None | 239 (26.3%) | 262 (30.2%) |
| Primary | 604 (66.4%) | 552 (63.7%) |
| Secondary or higher | 67 (7.4%) | 53 (6.1%) |
| Literate | 667 (73.3%) | 577 (66.6%) |
| Muslim | 672 (73.8%) | 686 (79.1%) |
| Ethnicity |  |  |
| Yao | 665 (73.1%) | 684 (78.9%) |
| Ngoni | 126 (13.8%) | 88 (10.1%) |
| Other | 119 (13.1%) | 95 (11.0%) |
| Resident in the last 2 months | 879 (96.6%) | 844 (97.3%) |
| Self-rated health status† |  |  |
| Very good | 416 (45.8%) | 328 (37.8%) |
| Good | 406 (44.7%) | 449 (51.8%) |
| Fair | 46 (5.1%) | 42 (4.8%) |
| Poor | 41 (4.5%) | 48 (5.5%) |
| Reported current sexual partner | 389 (42.7%) | 390 (45.0%) |
| Circumcised (for men) | 340 (87.9%) | 346 (90.8%) |

HIVST, HIV self-testing; SOC, standard of care.

*1 missing value in the SOC arm.

†1 missing value in the community-led HIVST arm.

## Process outcomes

Self-reported HIVST uptake was 74.7% (2,956/3,960) in the community-led HIVST arm, ranging from 68.5% in older men to 84.7% in young women (20 to 24 years), and 3.7% (145/3,920) in the SOC arm (Table 4 and Fig B in S2 Text). The proportion of participants aware of HIVST was 95.3% (3,771/3,960) and 32.4% (1,268/3,920) in the community-led HIVST and SOC arms, respectively.

Of 2,956 self-testers in the community-led HIVST arm, most obtained HIVST kits through primary distribution from community health groups (93.9%, *n* = 2,775). Only 4.4% (*n* = 130) received HIVST kits through secondary distribution from family members.

The majority of kits were obtained at the home of the participant (80.9%, *n* = 2,392) followed by the home of community health volunteers (7.4%, *n* = 220). Further, 10.4% (*n* = 306) reported no previous HIV testing and 2.4% (*n* = 70) reported an HIV-positive result, of whom

**Table 3. Primary and secondary outcomes by study arm.**

| | Community-led HIVST | | SOC | | Risk or mean difference (95% CI) | Adjusted risk or mean difference (95% CI)* | Risk ratio (95% CI) | Adjusted risk ratio (95% CI)* | |
|---|---|---|---|---|---|---|---|---|---|
| | n/N (%) | GM | n/N (%) | GM | *p*-value | *p*-value | *p*-value | *p*-value | k |
| **Primary outcome** | | | | | | | | | |
| Lifetime HIV testing among adolescents 15–19 years | 770/910 (84.6%) | 84.6% | 582/867 (67.1%) | 67.2% | 16.4% (7.8%–25.0%) <0.001 | 15.2% (7.5%–22.9%) <0.001 | 1.26 (1.11–1.43) <0.001 | 1.24 (1.11–1.39) <0.001 | 0.13 |
| Stratified by age group[†] | | | | | | | | | |
| 15–17 years | 320/400 (80.0%) | 79.5% | 219/384 (57.0%) | 54.3% | 22.5% (9.8%–35.3%) 0.001 | 21.5% (10.4%–32.6%) <0.001 | 1.47 (1.15–1.87) 0.003 | 1.44 (1.16–1.79) 0.002 | |
| 18–19 years | 450/510 (88.2%) | 88.0% | 363/483 (75.2%) | 76.0% | 11.5% (4.3%–18.7%) 0.003 | 10.8% (4.3%–17.3%) 0.002 | 1.16 (1.06–1.27) 0.003 | 1.15 (1.06–1.25) 0.002 | |
| Stratified by sex[‡] | | | | | | | | | |
| Male | 309/387 (79.8%) | 79.6% | 218/381 (57.2%) | 56.6% | 22.3% (11.9%–32.7%) <0.001 | 20.5% (10.7%–30.3%) <0.001 | 1.41 (1.19–1.66) <0.001 | 1.37 (1.17–1.6) <0.001 | |
| Female | 461/523 (88.1%) | 88.0% | 364/486 (74.9%) | 74.9% | 11.8% (2.6%–21.0%) 0.01 | 11.1% (2.8%–19.4%) 0.01 | 1.17 (1.04–1.33) 0.014 | 1.17 (1.04–1.31) 0.01 | |
| **Secondary outcomes** | | | | | | | | | |
| HIV testing in the last 3 months among adults ≥40 years[§] | 869/1,166 (74.5%) | 73.1% | 350/1,111 (31.5%) | 30.9% | 42.3% (34.7%–50.0%) <0.001 | 42.1% (34.9%–49.4%) <0.001 | 2.37 (2.0–2.79) <0.001 | 2.36 (2.01–2.77) <0.001 | 0.16 |
| HIV testing in the last 3 months among men | 1,177/1,577 (74.6%) | 73.8% | 507/1,495 (33.9%) | 33.3% | 40.8% (32.9%–48.6%) <0.001 | 40.2% (32.9%–47.4%) <0.001 | 2.22 (1.91–2.57) <0.001 | 2.19 (1.91–2.51) <0.001 | 0.17 |
| ART initiation per 100,000 population across 6 months[‖] | 136/44,543 (305.3) | 270.5 | 90/39,806 (226.1) | 207.3 | 72.3 (−36.2–180.8) 0.18 | | 1.31 (0.84–2.03) 0.23 | | 0.34 |
| Stratified by post-intervention period[¶] | | | | | | | | | |
| First 3 months | 83/44,543 (186.3) | 184.2 | 37/39,806 (93.0) | 97.5 | 97.7 (33.4–162.1) 0.004 | | 1.89 (1.21–2.95) 0.007 | | |
| Last 3 months | 53/44,543 (119.0) | 108.0 | 53/39,806 (133.2) | 122.8 | −10.7 (−80.5–59.2) 0.76 | | 0.88 (0.51–1.51) 0.63 | | |
| Knowledge of the preventive benefits of HIV treatment** | | 15.0 | | 14.7 | 0.3 (−0.6–1.3) 0.51 | 0.4 (−0.4–1.3) 0.29 | | | |
| HIV testing stigma[††] | | 7.4 | | 7.6 | −0.2 (−0.6–0.2) 0.36 | −0.2 (−0.5–0.2) 0.37 | | | |

ART, antiretroviral therapy; GM, geometric mean (of cluster-level proportions); HIVST, HIV self-testing; k, coefficient of variation in group village head-defined clusters; SOC, standard of care.

*Analysis adjusted for sex, age group, literacy, religion, ethnicity, and health status. Analysis among adolescents defines levels of age group as 16–17 years and 18–19 years. Analysis among adults ≥40 years defines levels of age group as 40–49 years and ≥50 years. Analysis among men adjusts for the same covariates except for sex.

[†]*p*-Value for interaction, *p* = 0.02.

[‡]*p*-Value for interaction, *p* = 0.06.

[§]Testing in the last 3 months was ascertained based on the most recent test date. If the month of the test date was not reported, we counted the test as being in the last 3 months if the test date was in 2018 for interview dates in 2018 or if the test date was in 2018 or 2019 for interview dates in 2019. 113 and 156 participants in the community-led HIVST and SOC arms, respectively, did not report month data. We conducted a sensitivity analyses where test dates with missing months were not counted as being in the last 3 months. Among adults ≥40 years, community-led HIVST: 72.0% (839/1,166), SOC: 27.0% (300/1,111); adjusted RD 43.7%, 95% CI 36.0%–51.5%; *p* < 0.001. Among men, community-led HIVST: 72.0% (1,135/1,577), SOC: 30.7% (459/1,495); adjusted RD 40.4%, 95% CI 32.7%–48.0%; *p* < 0.001.

[‖]Denominator for ART initiations is the estimated cluster population of adults ≥15 years, which was estimated using village and health facility registers and the proportion of adults reported in household enumeration.

[¶]Post hoc analysis. *p*-Value for interaction, *p* = 0.02.

**N = 1,925, with 30 missing values. Score is the sum of 5 questions using 5-point likert scale, with range of 5–25 (low to high knowledge).

[††]N = 1,929, with 26 missing values. Score is the sum of 6 questions using 3-point likert scale, with range of 3–18 (low to high stigma).

40.0% (*n* = 28) were newly identified and 11.4% (*n* = 8) were previously diagnosed and not on treatment. Self-reported ART initiation was 58.3% (21/36).

**Table 4. Fidelity to community-led HIV self-testing intervention.**

| | Community-led HIVST | SOC |
|---|---|---|
| | n (%) | n (%) |
| | (N = 3,960) | (N = 3,920) |
| Heard of self-testing* | 3,771 (95.3%) | 1,268 (32.4%) |
| Ever self-tested† | 2,956 (74.7%) | 145 (3.7%) |
| Self-tested in the last 3 months‡ | 2,919 (73.7%) | 128 (3.3%) |
| For most recent self-test: | (N = 2956) | |
| Self-test distributor§ | 2,775 (93.9%) | |
| Community health volunteer | 2,775 (93.9%) | |
| Family member | 130 (4.4%) | |
| Other | 49 (1.7%) | |
| Self-test collection location‖ | | |
| Home | 2,392 (80.9%) | |
| Home of community health volunteer | 220 (7.4%) | |
| Other | 343 (11.6%) | |
| First test ever | 306 (10.4%) | |
| Self-test result** | | |
| Positive†† | 70 (2.4%) | |
| Negative | 2,873 (97.4%) | |
| Invalid | 8 (0.3%) | |
| Harmed before or after self-testing‡‡ | 18 (0.6%) | |

HIVST, HIV self-testing; SOC, standard of care.

*1 missing value in the community-led HIVST arm and 1 missing value in the SOC arm.

†1 missing value in the community-led HIVST arm.

‡1 missing value in the community-led HIVST arm.

§2 missing values in the community-led HIVST arm.

‖1 missing value in the community-led HIVST arm.

¶1 missing value in the community-led HIVST arm.

**5 missing values in the community-led HIVST arm.

††40% (n = 28) were newly HIV–positive, 11.4% (n = 8) were previously diagnosed and not on treatment, 48.6% (n = 34) were previously diagnosed and on treatment. Of 36 HIV–positive and not on treatment, 58.3% (n = 21) initiated on antiretroviral therapy.

‡‡1 missing value in the community-led HIVST arm.

Adverse events related to HIVST were reported by 0.6% of participants (18/2,955) and classified by severity [24]. Reports included forced self-testing or results disclosure (moderate grade) and one case of physical harm (moderate to severe grade).

## Costs

Total provider cost of community-led HIVST was US$138,624, with a mean cost of US$ US$5.70 per HIVST kit distributed (Table B in S2 Text). Unit costs were US$241 per HIV–positive identified, US$602 per new HIV–positive identified, and US$468 per HIV–positive identified not on treatment.

## Discussion

Community-led delivery of 7-day HIVST campaigns linked to HIV treatment and prevention increased HIV testing in underserved subgroups. Lifetime testing increased by 15.2%

for adolescents, with more pronounced differences among younger adolescents and boys. Recent testing increased by 42.1% for older adults and by 40.2% for men. Mutual knowledge of HIV status between sexual partners also improved. Cumulative incidence of ART initiation per 100,000 population apparently increased 3 months post-intervention, with 186.3 residents treated in the community-led HIVST arm compared with 93.0 residents treated in the SOC arm. Difference in ART initiations between arms was not found for the predefined 6-month period. The intervention also achieved 74.7% HIVST uptake with limited adverse events and at US$5.70 per HIVST kit distributed. Our study provides evidence of an effective, safe, and affordable community strategy that rapidly achieved high impact and coverage at low cost and could be scaled in priority settings to meet and maintain HIV elimination goals.

To our knowledge, this is the first randomised trial to assess the impact of community-led delivery of HTS, which was recently enabled by the introduction of HIVST. This is also one of the few studies to report high coverage of HIV testing among subgroups with substantial undiagnosed HIV infection. Community participation has long been advocated as fundamental to primary healthcare and an approach that could increase coverage and efficiency of health programmes, improve outcomes, enhance the capacity of communities to address ill-health, and contribute to the sustainability of community health programmes [10]. We used participatory methods to engage established community health groups in designing and implementing HIVST campaigns adapted to their respective contexts. Our study builds on "top-down" community-based HIV testing and self-testing [18, 19] and "bottom-up" community-led demand creation for HIV services [12,15] by using a community-led HIVST model. Future iterations of this intervention could engage groups over time to provide repeat or multidisease services, including interventions to address malaria, tuberculosis, and other priority disease areas [25]. With the COVID-19 epidemic, community-led disease control programmes have potential to contribute to surveillance and early detection, reporting, and management. HIVST may also enable ongoing provision of HIV testing as routine HIV services are disrupted and reduce demand on healthcare workers to provide in-person HIV testing [26].

We found that community-led HIVST can lead to high coverage and effective targeting, with our study reporting substantially higher HIVST uptake from a community-led approach than a previous study of door-to-door HIVST [18]. Uptake was consistent across adolescents, older adults, and men. In contrast, vertical distribution of HIVST kits by community-based distribution agents achieved 42.5% uptake across a 12-month period in Malawi [18]. Uptake may be driven by successful context-informed planning, trust between community health groups and community members, and the value and novelty of HIVST among HIV-prevalent communities. The intervention also had minimal adverse events, alleviating safety concerns around decentralising management of HIVST implementation [16]. Further, our results may be applicable to high HIV prevalence settings in sub-Saharan Africa with similar cadres of community health workers and volunteers.

We showed increased lifetime and recent HIV testing in adolescents, especially younger adolescents and boys, older adults, and men, with prevalence of undiagnosed HIV disproportionately concentrated in these subgroups. Mutual knowledge of HIV status between sexual partners also increased. The intervention effect, while slightly lower than assumed for sample size calculations, was achieved against a SOC that included a high saturation of HIV services, with Mangochi a priority district for the Ministry of Health. Diagnosis of recent HIV infection is critical for HIV prevention, with our study reinforcing the importance of community strategies in reaching underserved subgroups [7]. Further, the impact attained within a short period of time makes community-led HIVST a promising candidate for national HIV programmes to consider for periodic implementation to reach underserved subgroups.

Community-led HIVST had an immediate impact on ART initiation 3 months post-intervention, though the effect diminished at 6 months. Population-level impact was measured even as the post-intervention survey reported that 1.2% of self-testers were newly HIV–positive or previously diagnosed but not on treatment, underscoring the potential for HIVST to influence ART demand. The intervention involved engaging government CHWs and health facilities to facilitate linkage to routine HIV services, likely contributing to successful referrals. However, self-reported ART initiation was 58.3% at follow-up. Optimising timely linkage to HIV treatment and prevention services is essential to maximise the health benefits from HIV testing and self-testing [27]. Neither VMMC nor preexposure prophylaxis were available at primary care level during the study, so we were unable to evaluate uptake of these services as intended. Further, despite providing training and materials to community volunteers on HIV prevention messages, the intervention did not improve knowledge of the preventive benefits of HIV treatment. The absence of effect may reflect insufficient discussion on the topic or difficulties conveying HIV risk reduction concepts.

Our analysis reported average cost of US$5.70 per HIVST kit distributed, which was lower than the cost of door-to-door HIVST models in nearby rural districts (2017 US$8.15) and urban Blantyre (2014 US$8.78) [28,29]. Average costs of community-based HTS in sub-Saharan Africa were similar [7]. Community health programmes are important for epidemic preparedness and management but can be costly to implement [9]. A community-led approach to HIVST is likely to realise significant economies of scale, with potential cost savings when community health groups are mobilised nationally and recurrently by Ministries of Health in non-research settings. Economies of scope can also lead to greater efficiency by implementing HIVST within a package of interventions addressing a broader set of conditions, including through the use of self-care products. Further, we found that the cost per HIV–positive identified through HIVST was US$241 to US$602, with modelling studies suggesting a cost-effectiveness threshold of US$315 per new HIV diagnosis [4]. Further cost reductions and uptake among undiagnosed, untreated HIV–positive persons or high HIV risk persons linking to HIV prevention would ensure greater probability of community-led HIVST as a cost-effective strategy.

Our study had multiple limitations. Due to the pragmatic nature of the intervention, there was some contamination of HIVST in the SOC arm, although reported events were nominal. The study design did not allow us to isolate the effects of specific intervention components, including the use of participatory methods and introduction of HIVST. Primary and secondary outcomes on HIV testing were self-reported and subject to misreporting due to recall or social desirability bias, including overreporting in the community-led HIVST arm following exposure to the intervention. We had a small number of clusters per arm and aimed to mimimise bias through randomisation of clusters, using restriction of factors likely to be associated with the outcome, and adjustment for imbalances between arms in individual characteristics. However, we did not measure primary and secondary outcomes in a baseline sample and adjust for baseline HIV testing. Our sampling frame may have included households that had better access to the HIVST intervention due to their location, with potential overestimation of the intervention effect. However, the effect size was relatively large, and our conclusions would have likely remained unchanged. The survey included fewer men than expected with almost one-fifth of eligible men not found. Implementation occurred within a controlled research setting as part of a mature HIVST programme that had been operating since 2015, potentially affecting the generalisability of our costs. Adverse events were reported in the survey, with follow-up to obtain details not feasible. We also did not evaluate accuracy of HIVST, which was previously shown to be high in rural settings when given optimised instructional materials and brief demonstrations [30].

Community-led delivery of 7-day HIVST campaigns linked to HIV treatment and prevention was effective in increasing HIV testing in adolescents, older adults, and men and mutual knowledge of HIV status between sexual partners. Population-level ART initiation apparently increased within a 3-month period but showed no difference at 6 months. Community-led delivery of HIVST was safe and associated with higher uptake and lower costs compared with previous evaluations of vertical community-based HIV testing and self-testing. Given evidence of high population impact and coverage rapidly realised at low cost, community-led HIVST shows much promise as an effective, safe, and affordable strategy, while entrusting communities with leading solutions for disease control. This approach could enable community HIV testing in high HIV prevalence settings and demonstrates potential for economies of scale and scope.

## Supporting information

**S1 CONSORT Checklist. Checklist of information to include when reporting a cluster-randomised trial.**
(DOCX)

**S1 Text. Statistical analysis plan.** Statistical analysis plan for primary and secondary outcomes.
(DOCX)

**S2 Text. Supporting Information.** Supporting text, tables, and figures.
(DOCX)

## Acknowledgments

We are grateful to the study participants; community partners and the Mangochi District Health Office for their involvement in the implementation of the study; the Department of HIV and AIDS at the Ministry of Health for their involvement in the study design; the Malawi-Liverpool-Wellcome Trust Clinical Research Programme and Population Services International Malawi team for their contributions to the study design and implementation; and the technical advisory group for their scientific input and guidance.

## Author Contributions

**Conceptualization:** Pitchaya P. Indravudh, Elizabeth L. Corbett.

**Data curation:** Pitchaya P. Indravudh, Rebecca Nzawa.

**Formal analysis:** Pitchaya P. Indravudh, Katherine Fielding, Fern Terris-Prestholt.

**Funding acquisition:** Karin Hatzold, Elizabeth L. Corbett.

**Investigation:** Pitchaya P. Indravudh, Katherine Fielding, Fern Terris-Prestholt, Elizabeth L. Corbett.

**Methodology:** Pitchaya P. Indravudh, Katherine Fielding, Moses K. Kumwenda, Richard Chilongosi, Nicola Desmond, Rose Nyirenda, Fern Terris-Prestholt, Elizabeth L. Corbett.

**Project administration:** Pitchaya P. Indravudh, Moses K. Kumwenda, Richard Chilongosi.

**Resources:** Pitchaya P. Indravudh, Moses K. Kumwenda, Richard Chilongosi.

**Software:** Pitchaya P. Indravudh, Rebecca Nzawa.

**Supervision:** Pitchaya P. Indravudh, Katherine Fielding, Rose Nyirenda, Melissa Neuman, Cheryl C. Johnson, Rachel Baggaley, Karin Hatzold, Fern Terris-Prestholt, Elizabeth L. Corbett.

**Validation:** Katherine Fielding, Fern Terris-Prestholt.

**Visualization:** Pitchaya P. Indravudh.

**Writing – original draft:** Pitchaya P. Indravudh.

**Writing – review & editing:** Katherine Fielding, Moses K. Kumwenda, Rebecca Nzawa, Richard Chilongosi, Nicola Desmond, Rose Nyirenda, Melissa Neuman, Cheryl C. Johnson, Rachel Baggaley, Karin Hatzold, Fern Terris-Prestholt, Elizabeth L. Corbett.

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
