## [Editor Report · Decision Letter 0]

5 Aug 2020

Dear Dr Indravudh, 

Thank you for submitting your manuscript entitled "Effect of community-led delivery of HIV self-testing on HIV testing and antiretroviral therapy initiation in Malawi: a cluster-randomised trial" for consideration by PLOS Medicine.

Your manuscript has now been evaluated by the PLOS Medicine editorial staff and I am writing to let you know that we would like to send your submission out for external assessment.

Kind regards,

Richard Turner, PhD

Senior editor, PLOS Medicine

rturner@plos.org

---

## [Decision Letter · Decision Letter 1]

9 Sep 2020

Dear Dr. Indravudh,

Thank you very much for submitting your manuscript "Effect of community-led delivery of HIV self-testing on HIV testing and antiretroviral therapy initiation in Malawi: a cluster-randomised trial" (PMEDICINE-D-20-03726R1) for consideration at PLOS Medicine. 

Your paper was evaluated by the editors and sent to independent reviewers, including a statistical reviewer. The reviews are appended at the bottom of this email and any accompanying reviewer attachments can be seen via the link below:

[LINK]

In light of these reviews, we will not be able to accept the manuscript for publication in the journal in its current form, but we would like to invite you to submit a revised version that addresses the reviewers' and editors' comments fully. You will appreciate that we cannot make a decision about publication until we have seen the revised manuscript and your response, and we expect to seek re-review by one or more of the reviewers. 

We hope to receive your revised manuscript by Sep 30 2020 11:59PM. Please email us (plosmedicine@plos.org) if you have any questions or concerns.

Please let me know if you have any questions, Otherwise, we look forward to receiving your revised manuscript in due course. 

Sincerely,

Richard Turner, PhD

rturner@plos.org

Please state briefly in the submission form where study data can be found - e.g., in supplementary files. 

Please quote aggregate demographic details for study participants in the abstract. 

Please add a new final sentence to the "Methods and findings" subsection of your abstract, quoting 2-3 of the study's main limitations. 

At line 69, please begin the sentence with "In this study, we found that ..." or similar. 

In the abstract and elsewhere, please quote p values alongside 95% CI, where available. 

After the abstract, we will need to ask you to add a new and accessible "author summary" section in non-identical prose. You may find it helpful to consult one or two recent research papers in PLOS Medicine to get a sense of the preferred style. 

Please remove the information on funding from the end of the main text. In the event of publication, this information will appear in the article metadata via entries in the submission form. 

Throughout the paper, please adapt reference call-outs to the following style: "... [5,6]." (i.e., no spaces within the square brackets). 

In reference 25, should that be "Taylor and Francis Group"?

Please adapt the attached CONSORT checklist so that individual items are referred to by section (e.g., "Methods") and paragraph numbers rather than by line or page numbers, as the latter generally change in the event of publication. Please refer to this in the Methods section (e.g., "See S1_CONSORT_Checklist").

Comments from the reviewers:

*** Reviewer #1: 

This article describes a cluster randomized trial in Malawi of community-led delivery of HIV self-testing with a focus on impact safety and cost. The study takes on an important topic in that lack of awareness of HIV, while decreasing overall, remains high in some sub-populations and testing is the first step towards increasing testing, increasing treatment initiation, decreasing community viral suppression and eventually decreasing incidence. The study was well conducted and the cluster randomized design was both appropriate and helps to increase the validity of the results. I commend the authors on their work and I am generally disposed toward supportingg publication. Still, there are some important limitation to the work which need to be dealt with prior to this study being ready for publication.

Below are specific comments about the manuscript.

Running the intervention in conjunction with local community groups is a very nice strength of the approach and will hopefully make it more sustainable.

I appreciate that the program provided support for travel for linkage to care. But is this a sustainable model? Can you talk more about this in the discussion.

The number of clusters was small, so the potential for bias remains and this is worth discussing. Still, the participants were well balanced so the amount of bias is hopefully small. What is missing is the most important thing, however, which is lifetime risk of testing before the intervention period. This should be in the discussion as well.

SOC was not well described. How much testing was already being done? It would seem that the effectiveness of the intervention would be in part driven by how much people already test so this is important. Also, for the intervention arm, did facility testing continue such that the intervention is really home and facility testing? Were any other testing campaigns being done at the time.

Could there have been any crossover/contamination? I suspect not, but I can't tell from the paper how close the populations were to each other and if the intervention were delivered in a way that someone could access them from outside.

To that point, I found details on delivery of the intervention to be lacking. You did a nice job describing the process by which the intervention was developed, but practically how were test kits delivered? They went door to door, but how often? Did they have a list of all homes to go to?

It isn't clear in the methods who the study population is. Everyone in the community? Just those who wanted testing? This is a bit clearer in the results, but the idea of an evaluation population isn't made very clear in the methods. Why wasn't everyone in the cluster included in the population? I assume for cost and time reasons, but then why wasn't a random sample taken?

I have some concern about the way the evaluation population was chosen as much as I understand it. Starting with the village head household could potentially lead to those houses with more access, high probability of uptake being chosen and those households farther away which may be less likely to uptake any HIV testing being less likely to participate.

The fact that outcomes are self-reported is a limitation. You mention this in the discussion but this needs more discussion. What is the likely direction and magnitude of this bias? I would think that those in the home testing group might have more incentive to say they tested due to the increased community mobilization, so it would bias away from the null.

In the statistical analysis, I was unclear on what this means "Risk/mean differences and

risk ratios were calculated from cluster-level summaries". Most importantly, what approach was used to adjust for the clustered design? I assume yes because you accounted for it in your sample size and using your data without cluster adjustment, I get narrower intervals, but I'm just not clear on how it was done.

In the sample size section, and this is not really very important, but I'm curious why your control arm had much higher lifetime testing than you anticipated.

In the sample size you specified a 20% difference. Was this the meaningful difference? I ask because you didn't meet this even though you did find a significant result.

*** Reviewer #2: 

Thank you for the opportunity review this manuscript on the effect of community-led delivery of HIV self-testing on case finding and ART initiation in Malawi. While some countries are approaching epidemic control, others are still lagging, and we must work in those countries to identify undiagnosed PLHIV. Particularly in the time of COVID-19, HIV self-testing represents a unique opportunity to provide a wide coverage of testing services to those who are unable to access testing sites due to stigma, transportation cost, lock-down restrictions, lack of awareness, etc. This manuscript provides important results indicating the efficacy of community HIV self-testing in Malawi, and will be important for moving case identification strategies forward.

Minor Revisions:

Methods: The inclusion/exclusion criteria is not entirely clear. Please better define how participants were selected and what the inclusion/exclusion criteria were for each study arm. 

Was counseling or social support provided to those who participated in HIV self-testing? 

Linkage to ART services for those who tested positive in SOC was mentioned, but was linkage to ART services also provided to those participating in HIV self-testing?

Statistical analysis: An alpha value was not denoted in the statistical analysis section, and therefore it is unclear what level of significance is being tested.

Discussion: In reference to the current atmosphere with COVID-19, the use of HIV self-testing as a way to limit clinic visits, reduce HRH testing demands, and reduce in-person testing services could be added to strengthen the argument for community-led HIV self-testing implementation and expansion.

*** Reviewer #3: 

This is a useful cluster-randomised trial on the effect of community-led delivery of HIV self-testing on HIV testing and antiretroviral therapy initiation in Malawi. However, there are quite a few issues needing attention.

1) Strictly speaking, The analyses are not intention to treat analyses. The definition of intention to treat analysis is 'once randomised always analysed'. However, around 10% of adolescents in each arm was missing for the primary analysis, and only 25% of those > 15 years old in each arm was available for the secondary analyses. These all happened after randomisation so could introduce potential biases. The current analyses are basically 'complete case analysis'.

2) The primary outcome is the lifetime HIV testing among adolescents (15-19 years) self-testing vs standard care. However, it's confusing on how this short-term intervention (7 days) over a period less than a year was related to lifetime HIV testing? Because it's more or less a cross-section survey on HIV testing. There was neither lifetime follow-up nor prior information before the trial on the testing. The second outcome on short-term measure might be more related to the effect of the intervention, but lifetime testing? Can authors justify the appropriateness of the primary outcome.

3) Sample size. Sample size was calculated on 20% efficacy in lifetime testing. Where did this 20% assumption come from? also what's justification of using coefficient of variation of 0.25 (assuming it is the intracluster correlation?)? This 0.25 seems too big and basically the sample size could not be reproduced. Detailed and clearer sample size calculation is needed with justifications of all the assumptions.

4) Statistical analysis. It said "Risk/mean differences and risk ratios were calculated from cluster-level summaries and compared between arms with t-test". This is not correct as t-test is for continuous variables. For categorical variables, need some other tests. It said "effect estimates were adjusted for sex and age group a priori and any imbalance between arms in adolescent covariates". However, can authors make it clear what exactly the stats models used for the adjusted analyses? What are the outcomes? What are the covariates?

5) Study design. Although the lifetime testing for adolescents is the primary outcome, it seems the investigators were more interested in households and adult residents in each cluster when designing the trial and randomising the clusters. It's a bit confusing what's rationale of the study design. Household first or adolescents first? It looks like the sample size calculation and identification of primary outcome is a post hoc decision as there are so many focus on secondary analyses which are not even powered. As we can see, the table 1 should be on the comparison of adolescents in two arms but it becomes about households? The current table 1 can be removed to supplementary information.

6) Randomisation. It only said "restricted randomisation" but exactly by what method? stratification, minimisation or blocking? Details are needed. Also there are professional software and CTU service around for the randomisation but two study PIs did this in a way that is not really understandable by professional statisticians. It should be simpler and more straightforward than what's written in the paper.

***

[LINK]

---

## [Decision Letter · Decision Letter 2]

13 Nov 2020

Dear Dr. Indravudh,

Thank you very much for submitting your revised manuscript "Effect of community-led delivery of HIV self-testing on HIV testing and antiretroviral therapy initiation in Malawi: a cluster-randomised trial" (PMEDICINE-D-20-03726R2) for consideration at PLOS Medicine. 

Your paper was again evaluated by our independent reviewers, including a statistical reviewer. The reviews are appended at the bottom of this email and any accompanying reviewer attachments can be seen via the link below:

[LINK]

We will not yet be able to accept the manuscript for publication in the journal in its current form, but we would like to invite you to submit a further revised version that addresses the reviewers' and editors' comments fully. You will note that we cannot make a decision about publication until we have seen the revised manuscript and your response, and we may seek re-review by one or more of the reviewers. 

We hope to receive your revised manuscript by Dec 04 2020 11:59PM. Please email us (plosmedicine@plos.org) if you have any questions or concerns.

Please let me know if you have any questions. Otherwise, we look forward to receiving your revised manuscript soon. 

Sincerely,

Richard Turner, PhD

rturner@plos.org

Please finalize the arrangements for data release. 

We suggest amending the text at line 34 to "... HIV infection ... in key groups including ...".

Please revisit the cumulative incidence of ART initiation, risk difference "73.2%" quoted at line 68. This appears to be quoted as "72.7%" in table 2. 

Also at line 68, please adapt the wording to make it clear what "Differences were observed ..." refers to, perhaps by linking this to the previous sentence. Please adapt the wording if this is a post-hoc analysis. Again, please make sure that the numbers quoted can be reconciled with those at line 104 and in table 2. 

At line 104, we ask you to adapt the wording to "apparently increased" or similar, as we believe you are quoting a secondary outcome finding where the risk of type 1 error is elevated. Please make a similar change at line 462.

At line 547, we suggest using the word "contamination" rather than "spillover". 

Comments from the reviewers:

*** Reviewer #1: 

Overall I am happy with the responses except for:

The number of clusters was small, so the potential for bias remains and this is worth discussing. Still, the participants were well balanced so the amount of bias is hopefully small. What is missing is the most important thing, however, which is lifetime risk of

testing before the intervention period. This should be in the discussion as well.

We have provided additional details on the sample size calculations (pg. 14), which are based on standard methods outlined by Hayes and Moulton for cluster-randomised trials [1]. Although the number of clusters was small, the trial was designed with sufficient power to

measure relevant differences between arms in the primary and secondary outcomes. Our study did not measure lifetime risk of testing through a baseline sample; we have expanded on this as a limitation in the discussion (pg. 26).

My point is not about power, my point is about bias. Small numbers of clusters means that the chance of lack of exchangeable populations increases and this is an important limitation of the design that needs to be discussed.

*** Reviewer #2: 

Thank you for the opportunity re-review this paper looking at the impact of community-led HIVST on case identification and initiation on treatment. This is an important and relevant study that shows the effectiveness of HIVST in improving case identification, treatment initiation, viral suppression, and eventually reducing incidence. The author has addressed all of my and my fellow reviewer's comments in this paper with sufficient justification, and I therefore recommend this paper be accepted for publication.

*** Reviewer #3: 

Many thanks authors for their great effort to improve the manuscript. The authors have addressed most of my concerns and comments adequately. However, there is still one remaining issue needing attention. I am still not satisfied/convinced with the answer for using the lifetime HIV testing among adolescents (15-19 years) as primary outcome of the study. I can understand it from practical point of view but couldn't understand what's research hypothesis and potential causal relationship between the short-term intervention and lifetime HIV testing, direct or indirect? any evidence (similar studies) from the existing literature? The secondary outcomes are more straightforward and making sense so it's fine.

***

[LINK]

---

## [Decision Letter · Decision Letter 3]

17 Dec 2020

Dear Dr. Indravudh,

Thank you very much for re-submitting your manuscript "Effect of community-led delivery of HIV self-testing on HIV testing and antiretroviral therapy initiation in Malawi: a cluster-randomised trial" (PMEDICINE-D-20-03726R3) for consideration at PLOS Medicine.

I have discussed the paper with editorial colleagues and our academic editor, and it was also seen again by one reviewer. I am pleased to tell you that, provided the remaining editorial and production issues are fully dealt with, we expect to be able to accept the paper for publication in the journal.

[LINK]

We hope to receive your revised manuscript after the holidays. Please email us (plosmedicine@plos.org) if you have any questions or concerns.

Please let me know if you have any questions. Otherwise, we look forward to receiving the revised manuscript soon.   

Sincerely,

Richard Turner, PhD

rturner@plos.org

Requests from Editors:

Please finalize the arrangements for study data deposition. 

At line 64, please amend the text to "the proportions ... were ...". 

Please revisit the sentence at lines 69-70, where it seems that a few additional words would be helpful to indicate that the difference is in favour of the HIVST group. 

At line 71, please make that "few adverse events".

We note that reference 16 contains some superfluous information about competing interests - please read through the reference list to ensure that this and any other material not needed is removed. 

Please move table S1 to the main paper and adjust the table numbering as appropriate. 

Please break up the supplementary material document into individual files, so that specific items and files are referred to in the text (e.g., "See S2_Figure"). 

The attached published trial protocol can be removed and simply referred to as a reference. 

Comments from Reviewers:

*** Reviewer #3: 

The authors have addressed my question very well. I am satisfied with the response and revision. No further issues needing attention.

***

[LINK]

---

## [Editor Report · Decision Letter 4]

4 Apr 2021

Dear Dr Indravudh, 

On behalf of my colleagues and our Academic Editor, Dr Fox, I am pleased to inform you that we have agreed to publish your manuscript "Effect of community-led delivery of HIV self-testing on HIV testing and antiretroviral therapy initiation in Malawi: a cluster-randomised trial" (PMEDICINE-D-20-03726R4) in PLOS Medicine.

Prior to final acceptance, please add a few words around line 160 to note that the trial is reported according to CONSORT.

PRESS

Sincerely, 

Richard Turner, PhD 

rturner@plos.org